# Swarm Intelligence in Geo-Localization: A Multi-Agent Large Vision-Language Model Collaborative Framework

## Abstract

Visual geo-localization demands in-depth knowledge and advanced reasoning skills to associate images with real-world geographic locations precisely. In general, traditional methods based on data-matching are hindered by the impracticality of storing adequate visual records of global landmarks. Recently, Large Vision-Language Models (LVLMs) have demonstrated the capability of geo-localization through Visual Question Answering (VQA), enabling a solution that does not require external geo-tagged image records. However, the performance of a single LVLM is still limited by its intrinsic knowledge and reasoning capabilities. Along this line, in this paper, we introduce a novel visual geo-localization framework called smileGeo that integrates the inherent knowledge of multiple LVLM agents via inter-agent communication to achieve effective geo-localization of images. Furthermore, our framework employs a dynamic learning strategy to optimize the communication patterns among agents, reducing unnecessary discussions among agents and improving the efficiency of the framework. To validate the effectiveness of the proposed framework, we construct GeoGlobe, a novel dataset for visual geo-localization tasks. Extensive testing on the dataset demonstrates that our approach significantly outperforms state-of-the-art methods. The source code is available at https://anonymous.4open.science/r/ViusalGeoLocalization-F8F5/ and the dataset will also be released after the paper is accepted.

## 1 Introduction

Visual geo-localization, referred to the task of estimating geographical identification for a given image, is vital in various fields such as human mobility analysis [1, 2, 3, 4, 5] and robotic navigation [6, 7, 8, 9, 10, 11]. In general, accurate visual geo-localization without the help of any localization equipment (*e.g.,* GPS sensors) is a complex task that requires abundant geospatial knowledge and strong reasoning capabilities. Traditional methods [12, 13, 14, 15] typically formulate it as an image retrieval problem where to geo-localize the given image by retrieving similar images with known geographical locations. Thus, their effectiveness is limited by the scope and quality of the geo-tagged image records.

Recently, the success of Large Vision-Language Models (LVLMs) has enabled Visual Question Answering (VQA) to become a unified paradigm for multi-modal problems [16, 17], providing a novel solution for visual geo-localization without the need for external geo-tagged image records. However, the performance of a single LVLM on the geo-localization task is still limited by its inherent geospatial knowledge and reasoning capabilities. Along this line, in this paper, we introduce a novel multi-agent framework, named **s**war**m i**ntel**lige**nce **Geo**-localization (**smileGeo**), which aims to adaptively integrate the inherent knowledge and reasoning capabilities of multiple LVLMs

to effectively and efficiently geo-localize images. Specifically, for a given image, the framework initially elects $K$ suitable LVLM agents as answer agents for initial location analysis. Then, each answer agent chooses several review agents via an adaptive social network, which imitates the collaborative relationships between agents with a target on the visual geo-localization task, to discuss and share their knowledge for refining its location analysis. Finally, our framework conducts free discussion among all of the answer agents to reach a consensus. Besides, we also design a novel dynamic learning strategy to optimize the election mechanism along with the adaptive collaboration social network of agents. We hope that by the effectiveness of the election mechanism and the review mechanism, our framework can discover the mode of communication among agents, thereby enhancing geo-localization performance through multi-agent collaboration while minimizing unnecessary discussions. In summary, our contributions are demonstrated as follows:

- A novel swarm intelligence geo-localization framework, smileGeo, is proposed to adaptively integrate the inherent knowledge and reasoning capability of multiple LVLMs through discussion for visual geo-localization tasks.

- A dynamic learning strategy is introduced to discover the most appropriate discussion mode among LVLM agents for enhancing the effectiveness and efficiency of the framework.

- A new visual geo-localization dataset named GeoGlobe[1] is collected, containing a wide variety of images globally. The diversity and richness of GeoGlobe allow us to evaluate the performance of different models more accurately. Moreover, extensive experiments demonstrate our competitive performance compared to state-of-the-art methods.

The remainder of this paper is organized as follows: Section 2 discusses the related literature. In Section 3, the proposed framework is introduced. Section 4 provides the performance evaluation, and Section 5 concludes the paper.

## 2   Related Work

**Visual Geo-localization**. Recent research in visual geo-localization, commonly referred to as geo-tagging, primarily focuses on developing image retrieval systems to address this challenge [3, 18, 19, 20, 21, 22]. These systems utilize learned embeddings generated by a feature extraction backbone, which includes an aggregation or pooling mechanism [23, 24, 25, 26]. However, the applicability of these retrieval systems to globally geo-localize landmarks or natural attractions is often limited by the constraints of the available database knowledge and the restrictions imposed by national or regional geo-data protection laws. Alternatively, some studies treat visual geo-localization as a classification problem [27, 28, 29, 30]. These approaches posit that two images from the same geographical region, despite depicting different scenes, typically share common semantic features. Practically, these methods organize the geographical area into discrete cells and categorize the image database accordingly. This cell-based categorization facilitates scaling the problem globally, provided the number of categories remains manageable. However, while the number of countries globally remains relatively constant, accurately enumerating cities in real-time at a global scale is challenging due to frequent administrative changes, such as city reorganizations or mergers, which reflect shifts in national policies. Additionally, in the context of globalization, this strategy has inherent limitations. The recent advent of LVLMs offers promising compensatory mechanisms for the deficiencies observed in traditional geo-localization methodologies, making the exploration of LVLM-based approaches significantly relevant in current research.

**Multi-agent Framework for LLM/LVLMs**. LLM/LVLM agents have demonstrated the potential to act like human [31, 32, 33], and a large number of studies have focused on developing robust architectures for collaborative LLM/LVLM agents [34, 35, 36, 37, 38]. These architectures enable each LLM/LVLM agent that endows with unique capabilities to engage in debates or discussions. For instance, [34] proposes an approach to aggregate multiple LLM/LVLM responses by generating candidate responses from various LLM/LVLM in a single round and employing pairwise ranking to synthesize the most effective response. While some studies [34] utilize a static architecture potentially limiting the performance and generalization of LLM/LVLM, others like [38] have implemented dynamic interaction architectures that adjust according to the query and incorporate user feedback.

---

[1]Because GeoGlobe is relatively large (about 32GB), we are unable to provide it as an attachment during the double-blind review stage. We will publish it once the paper is accepted.

Recent advancements also demonstrate the augmentation of LLM/LVLM as autonomous agents capable of utilizing external tools to address challenges in interactive settings. These techniques include retrieval augmentation [39, 40, 41], mathematical tools [40, 42, 43], and code interpreters [44, 45]. With these capabilities, LLM/LVLMs are well-suited for various tasks, especially for geo-localization. However, most LLM/LVLM agent frameworks mandate participation from all agents in at least one interaction round, leading to significant computational overhead. To address this issue, our framework introduces a dynamic learning strategy electing only a small number of agents to geo-localize different images, which significantly enhances the efficiency of LLM/LVLM agents by reducing unnecessary interactions.

## 3  Methodology

In this section, we first present the overall framework and then introduce each part of smileGeo in detail for geo-localization tasks.

### 3.1  Model Overview

In this paper, we denote the social network of LVLM agents by $\mathcal{G}$, where $\mathcal{G} = \{\mathcal{V}, \mathcal{E}\}$. $\mathcal{V}$ stands for the agent set and $\mathcal{E}$ presents the edge set. Each agent $v_i \in \mathcal{V}, i \in [N]$ is an LVLM, which is pre-trained by massive vision-language data and can infer the possible location $\boldsymbol{Y}$ of a given image $\boldsymbol{X}$. Besides, each edge $e_{ij} \in \mathcal{E}, i, j \in [N]$ is the connection weighted by the improvement effect of agent $v_i$ to agent $v_j$ via discussion regarding the geo-localization performance.

As illustrated in Figure 1, smileGeo contains the process of the review mechanism in agent discussions along with a dynamic learning strategy of agent social networks:

The review mechanism in agent discussions is a 3-stage anonymous collaboration approach to allow LVLM agents to reach a consensus via discussion. In the first stage, for a given image $\boldsymbol{X}$, our framework elects the most suitable $K$ agents as answer agents by agent election probability $\boldsymbol{Lst}$. In the second stage, these answer agents respectively select $R$ review agents by the adaptive collaboration social network $\boldsymbol{A}$ to refine their answer via discussion. Finally, our framework facilitates consensus among all agents through open discussion to reach a final answer. Both $\boldsymbol{Lst}$ and $\boldsymbol{A}$ are analyzed from the given image $\boldsymbol{X}$, allowing our framework to minimize unnecessary discussions, thereby significantly enhancing its efficiency while maintaining its accuracy. Moreover, the multi-stage discussion facilitates communication among agents, maximizing the integration of their knowledge and reasoning abilities to generate an accurate response $\boldsymbol{Y}$.

To get $\boldsymbol{Lst}$ and $\boldsymbol{A}$, we specifically design a dynamic learning module, which initially deploys the encoder component of a pre-trained image variational autoencoder (VAE) to extract features from the given image $\boldsymbol{X}$. The extracted features, combined with agent embeddings $\boldsymbol{Emb}$, are employed to determine the suitability of agents *w.r.t.* $\boldsymbol{Lst}$ for agent discussions and predict agent collaboration connections $\boldsymbol{A}$ in the geo-localization task.

### 3.2  Review Mechanism in Agent Discussions

LLM/LVLM have demonstrated remarkable capabilities in complicated tasks and some pioneering works have further proven that the performances can be further enhanced by ensembling multiple LLM/LVLM agents. Thus, to improve the geo-localization capability of LVLMs, we propose a cooperation framework to effectively integrate the diverse knowledge and reasoning abilities of multiple LVLMs. Inspired by the fact that community review mechanisms can improve the quality of manuscripts, an iterative 3-stage anonymous reviewing mechanism is proposed for helping agents share knowledge and reasoning capability with each other through their collaboration social network: i) answer agent election & answering, ii) review agent selection & reviewing, and iii) final answer conclusion.

**Stage 1: Answer Agent Election & Answering**

Initially, we select $K$ agents with the highest agent election probabilities $\boldsymbol{Lst}$ as answer agents and let them geo-localize independently as the preliminary step for further discussion. By initiating the discussion with a limited number of agents, we aim to reduce potential chaos and maintain the efficiency of our framework as the number of participating agents increases.

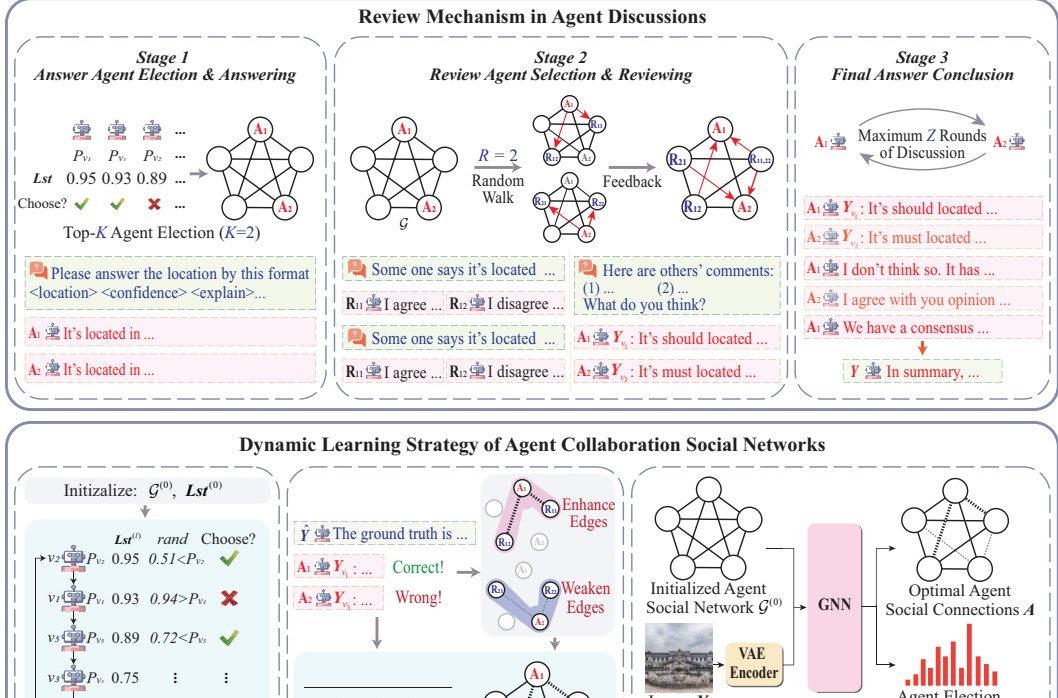

Figure 1: The framework overview of smileGeo. It contains the process of review mechanism in agent discussions along with a dynamic learning strategy of agent collaboration social networks. The first part deploys a review mechanism for LVLMs to discuss and share their knowledge anonymously, which could enhance the overall performance of geo-localization tasks. The second one mainly utilizes the GNN-based learning module to improve efficiency by reducing unnecessary discussions among agents while showing the process of updating the agent collaboration social network during the training process.

After the answer agents are elected, we send the image $X$ to all answer agents and let them give the primary analysis. Each answer must contain three parts: one location (city, country, and so on), one confidence (a percentage number), and a detailed explanation.

**Stage 2: Review Agent Selection & Reviewing**

In this stage, for each answer agent, we choose $R$ review agents by performing a transfer-probability-based random walk on the agent collaboration social network $\mathcal{G}$ for answer reviewing. The transfer probability $p(v_i, v_j)$ from node $v_i$ to node $v_j$ can be calculated as follows:

$$
p(v_i, v_j) = \begin{cases} \frac{\boldsymbol{A}_{ij}}{\sum_{k \in \mathcal{N}(v_i)} \boldsymbol{A}_{ik}}, & \text{if } e_{ij} \in \mathcal{E} \\ 0, & \text{otherwise} \end{cases} \tag{1}
$$

where $\mathcal{N}(v_i)$ is the 1-hop neighbor node set of node $v_i$.

For each selected review agent, it reviews the results as well as the explanations generated by the corresponding answer agent and gives its own comments. After that, each answer agent would summarize their preliminary analysis and the feedback from all of its review agents to get the final answer, which must include three parts as well: one location, one confidence, and an explain.

**Stage 3: Final Answer Conclusion**

In the previous stage, each answer agent produces a refined result based on feedback. When $K > 1$ in Stage 1, the proposed framework generates multiple independent results, which may not be consistent.

However, we aim to provide a definitive answer rather than multiple options for people to choose from. To address this, we allow up to $Z$ rounds of free discussion among those answer agents to reach a unified answer:

First, we maintain a global dialog history list, $diag$, recording all replies agents respond. In addition, discussions are executed asynchronously, which means that any answer agent can always reply based on the latest $diag$, and replies would be added to the end of $diag$ as soon as they are posted. Each answer agent is allowed to speak only once in each discussion round, and after $Z$ rounds of free discussion, we determine the final result using a minority-majority approach, *i.e.,* we choose the reply with the most agreement as the final conclusion. If all agents reach a consensus, we early stop this stage and adopt the consensus answer as the final answer. If none of any consensus is reached, we only select the reply of the first answer agent elected from Stage 1 as the final result.

### 3.3 Dynamic Learning Strategy of Agent Collaboration Social Networks

In our framework, choosing the appropriate answer agents and review agents for knowledge sharing and discussion is vital to its effectiveness and efficiency. Therefore, we propose a dynamic learning strategy to optimize them. Specifically, for each training sample, *i.e.,* a geo-tagged image, we would first estimate the optimal answer agent election probability $\hat{Lst}$ and the optimal collaboration social network of agent $\hat{\mathcal{G}}$ by its actual location. Then we train an attention-based graph neural network, which aims to predict $Lst$ and $\mathcal{G}$, by such estimated ground truth.

To estimate the optimal $\hat{Lst}$ and $\hat{A}$ for agents to geo-localize image $X$, we first initialize the agent social network $\mathcal{G}^{(0)}$ by a fully connected graph with the agent set $\mathcal{V}$. Besides, we initialize the agent election probability $Lst^{(0)} = [0.5, 0.5, \cdots]$, with all agents having $50\%$ probability of being chose as answer agents.

Then, we iteratively conduct our 3-stage discussion framework to get the prediction answer. $Lst^{(l)}$ and $\mathcal{G}^{(l)}$ is updated at the end of each round $l \in L$ by comparing the answers $Y_{v_i}^{(l)}$ from each answer agent with the ground truth $\hat{Y}$.

After $L$ rounds of agent discussions, the updated agent election probability for an image $X$, $\hat{Lst} := Lst^{(L)}(X) = [P_{v_1}^{(L)}, P_{v_2}^{(L)}, \cdots, P_{v_N}^{(L)}]$, determines whether an agent $v_i$ gives the correct/wrong answers $Y_{vi}^{(L)}$ by comparing it with the ground truth $\hat{Y}$. Here, the definition of $P_{v_i}^{(l)}$ of agent $v_i$ at round $l$ is as follows:

$$P_{v_i}^{(l)} := \begin{cases} 0, & \text{if } \mathcal{D}(\hat{Y}, Y_{v_i}^{(l)}) > th \\ 1, & \text{if } \mathcal{D}(\hat{Y}, Y_{v_i}^{(l)}) \leq th \\ \frac{1}{2}, & \text{if } v_i \text{ did not participate in the discussion} \end{cases} \tag{2}$$

where $th$ is a pre-defined threshold for determining whether the predicted location is close enough to the actual location. In the distance function $\mathcal{D}(\cdot)$, we first deploy geocoding to convert natural language into location intervals in a Web Mercator coordinate system (WGS84) by utilizing OSM APIs, and then compute the shortest distance between two two location intervals.

Please note that, rather than electing the top-$K$ answer agents in each round, we choose each agent with probability $P_{v_i}$ during the training period to ensure that every agent has the opportunity to participate in the discussion for more accurate estimation, as shown at the left part of the dynamic learning strategy module of agent collaboration social networks in Figure 1.

In addition, the agent collaboration social network would also be updated by comparing the actual location with the generated answer of each answer agent at the same time. For $l$-th round, we strengthen the link between the correctly answered agent and the corresponding review agents while weakening the link between the incorrectly answered agent and the corresponding review agents:

$$\hat{A}_{ij} := A_{ij}^{(l)}(X) = \begin{cases} \frac{tt+1}{2tt} A_{ij}^{(l-1)}(X), & \text{if agent } v_i \text{ answers correctly} \\ \frac{2tt-1}{2tt} A_{ij}^{(l-1)}(X), & \text{if agent } v_i \text{ answers incorrectly} \end{cases} \tag{3}$$

where $\boldsymbol{A}_{ij}^{(l-1)}(\boldsymbol{X})$ is the weight of the connection between answer agent $v_i$ and review agent $v_j$ at round $l-1$ when geo-locating image $\boldsymbol{X}$, $\boldsymbol{A}_{ij}^{(0)}(\boldsymbol{X}) = 1, i \neq j, \boldsymbol{A}_{ii}^{(0)}(\boldsymbol{X}) = 0, i, j \in [N]$, $tt$ is the number of consecutive times an agent has answered correctly, which is used to attenuate the connection weights when updating them, preventing the performance of an agent on a certain portion of the continuous dataset from interfering with the model's evaluation of the current agent's performance on the entire dataset.

Then, we try to learn an attention-based graph neural network to predict the corresponding optimal agent election probability $\boldsymbol{Lst} = h(\boldsymbol{X}, \mathcal{G}|\Theta)$ and the optimal agent collaboration connections $\boldsymbol{A} = f(\boldsymbol{X}, \mathcal{V}|\Theta)$:

$$
\begin{aligned}
\boldsymbol{A} &= \mathrm{Att}_{\mathrm{GNN}}(\boldsymbol{Fea}, \boldsymbol{Fea}, \boldsymbol{1}) \\
&= \mathrm{softmax}\left(\frac{\boldsymbol{Fea} \cdot \boldsymbol{Fea}^\top}{\sqrt{d_k}}\right)\boldsymbol{1}, \\
\boldsymbol{Lst} &= \sigma'\left(\mathrm{Linear}\left(\mathrm{Flatten}\left(\sigma\left(\boldsymbol{A} \cdot \boldsymbol{Fea} \cdot \boldsymbol{W}\right)\right)\right)\right), \\
\boldsymbol{Fea} &= \mathrm{Linear}\left(\boldsymbol{Emb} + \mathrm{VAE}_{\mathrm{Enc}}(\boldsymbol{X})\right),
\end{aligned}
\tag{4}
$$

where $\boldsymbol{W}, \boldsymbol{Emb} \in \Theta$ are two learnable parameters, $\boldsymbol{Emb} := [\boldsymbol{Emb}_{v_1}, \boldsymbol{Emb}_{v_2}, \cdots]^\top$ is the agent embedding and $\boldsymbol{W}$ is the weight matrix, $\sigma(\cdot)$ is the LeakyReLU function, $\sigma'(\cdot)$ is the Sigmoid function, $\mathrm{VAE}_{\mathrm{Enc}}(\cdot)$ is the encoder of the image VAE that compresses and maps the image data into the latent space. It is used to align the image features with the agent embedding, and $d_k$ is the dimension of the $\boldsymbol{Fea}$. Our learning target can be formalized as:

$$
\arg\min_{\Theta} \sum_i^N \mathcal{D}(\hat{\boldsymbol{Y}}, \boldsymbol{Y}_{v_i})\mathbb{1}(v_i \text{ gives an answer}) + \mathrm{MSE}(\hat{\boldsymbol{Lst}}, \boldsymbol{Lst}) + \mathrm{MSE}(\hat{\boldsymbol{A}}, \boldsymbol{A}),
\tag{5}
$$

where $\mathcal{D}(\cdot)$ denotes the distance between the places an LVLM agent answered and the ground truth, $\mathbb{1}(\cdot)$ is the indicator function, $\boldsymbol{Y}_{v_i} := \boldsymbol{Y}_{v_i}^{(L)} = g_{v_i}(\boldsymbol{X}, \boldsymbol{Y}_{v_j}^{(L-1)})$, $g_{v_i}(\cdot)$ represent the LVLM agent $v_i$ with fixed parameters and $\boldsymbol{Y}_{v_i}^{(0)} = g_{v_i}(\boldsymbol{X})$ is the answer that LVLM agent $v_i$ generates at the initial stage of discussion.

# 4 Experiments

To evaluate the performance of our framework, we conducted experiments on the real-world dataset that was gathered from the Internet to answer the following research questions:

• **RQ1**: Can smileGeo outperform state-of-the-art methods in open-ended geo-localization tasks?

• **RQ2**: Are LVLM agents with diverse knowledge and reasoning abilities more suitable for building a collaboration social network of agents?

• **RQ3**: How does the setting of hyperparameters affect the performance of smileGeo?

## 4.1 Experiment Setup

**Datasets**. In this paper, we newly construct a geo-localization dataset named GeoGlobe. It contains a variety of man-made landmarks or natural attractions from nearly 150 countries with different cultural and regional styles. The diversity and richness of GeoGlobe allow us to evaluate the performance of different models more accurately. More details can be found in Appendix B.

**Implementation Details**. We select both open-source and close-source LVLMs with different scales trained by different datasets as agents in the proposed framework. As for the open-source LVLMs, we utilize several open-source fine-tuned LVLMs: Infi-MM[2], Qwen-VL[3], vip–llava–7b&13b[4], llava–

---

[2]https://huggingface.co/Infi-MM/infimm-zephyr
[3]https://huggingface.co/Qwen/Qwen-VL
[4]https://huggingface.co/llava-hf/vip-llava-xxx

1.5–7b–base&mistral&vicuna[5], llava–1.6–7b&13b&34b–mistral&vicuna[6], CogVLM[7]. As for the closed-source LVLMs, we chose the models provided by three of the most famous companies in the world: Claude–3–opus[8], GPT–4V[9], and Gemini–1.5–pro [10]. Besides, 99% of images (about 290,000 samples) from the original dataset are randomly chosen as training samples. For the open-world geolocation problem, we construct the test dataset using approximately 4,000 samples, of which nearly 66.67% samples reflected different locations not present in the training dataset. More details about the deployment of smileGeo and the related parameter settings can be found in Appendix C.

**Baselines**. In this work, we compare the proposed framework with three kinds of baselines: single LVLMs, LLM/LVLM-based multi-agent frameworks, and image retrieval approaches. Firstly, we use each LVLM alone as an agent directly for the geo-localization task and compute the performance of these single LVLMs under the same dataset. In addition, we experiment with multi-agent collaborative frameworks, including LLM-Blender [34], PHP [35], Reflexion [36], LLM Debate [37], and DyLAN [38]. Finally, several state-of-the-art image retrieval approaches, including NetVLAD [3], GeM [26], and CosPlace [46], are also used to be part of the baselines. We set the training dataset as the geo-tagged image database of each image retrieval system and use images in the test dataset for the retrieval system to generate answers.

**Evaluation Metrics**. We use *Accuracy* (Acc) to evaluate the performance: $Accuracy = \frac{N_{correct}}{N_{total}}$, where $N_{correct}$ is the number of samples that the proposed framework correctly geo-localizes, and $N_{total}$ refers to the total number of testing samples.

In this paper, we first geo-encode the answers with the ground truth, *i.e.,* we transform the addresses described through natural language into latitude-longitude coordinates. Then, we calculate the distance between the two coordinates. When the distance between the two coordinates is less than $th = 50km$ (city-level), we consider the answer of the framework to be correct.

## 4.2 Performance Comparison

We divide the baseline comparison experiment into three parts: i) comparison with single LVLMs, ii) comparison with LLM/LVLM-based agent frameworks, and iii) comparison with image retrieval systems.

Table 1: Results of different single LVLM baselines.

| | Without Web Searching | | | With Web Searching | | |
|---|---|---|---|---|---|---|
| | **Natural** | **ManMade** | **Overall** | **Natural** | **ManMade** | **Overall** |
| Infi-MM | 19.2547 | 21.4133 | 20.9883 | 0.9938 | 0.3351 | 0.4648 |
| Qwen-VL | 42.4845 | 37.4657 | 38.4540 | 4.9689 | 11.2093 | 9.9804 |
| vip-llava-13b | 20.6211 | 15.4127 | 16.4384 | 8.323 | 4.3558 | 5.137 |
| vip-llava-7b | 21.9876 | 18.4892 | 19.1781 | 31.9255 | 56.5032 | 51.6634 |
| llava-1.5-7b | 17.3913 | 16.3265 | 16.5362 | 27.205 | 47.2129 | 43.273 |
| llava-1.6-7b-mistral | 0.3727 | 0.0914 | 0.1468 | 0.8696 | 2.1627 | 1.908 |
| llava-1.6-7b-vicuna | 2.2360 | 2.0713 | 2.1037 | 6.9565 | 15.8696 | 14.1145 |
| llava-1.6-13b | 10.4348 | 8.8943 | 9.1977 | 12.1739 | 28.2668 | 25.0978 |
| llava-1.6-34b | 10.3106 | 9.1379 | 9.3689 | 52.795 | 77.1855 | 72.3826 |
| CogVLM | 7.7019 | 7.5845 | 7.6076 | 6.8323 | 10.3564 | 9.6624 |
| claude-3-opus | 22.06 | 37.38 | 16.5468 | 33.0435 | 40.7125 | 39.2027 |
| GPT-4V | 27.5776 | 35.3443 | 33.8145 | 61.9876 | 87.6028 | 82.5587 |
| Gemini-1.5-pro | 55.6522 | 60.3107 | 59.3933 | 62.2360 | 82.8206 | 78.7671 |
| **smileGeo** | **58.6111** | **64.3968** | **63.2730** | **78.0448** | **87.0069** | **85.2630** |

Bold indicates the statistically significant improvements
(*i.e.,* two-sided t-test with $p < 0.05$) over the best baseline.

---

[5]https://huggingface.co/llava-hf/llava-1.5-xxx

[6]https://huggingface.co/liuhaotian/llava-v1.6-xxx

[7]https://github.com/THUDM/CogVLM

[8]https://anthropic.com/

[9]https://openai.com/

[10]https://gemini.google.com/

Firstly, the performance of all single LVLM baselines is shown in Table 1, in terms of the metric Acc. The data in Table 1 indicate that open-source LVLMs with diverse knowledge and reasoning capabilities exhibit significant variations, particularly in geo-localization tasks. This may be due to the difference in the overlap between the pre-training datasets used by different LVLMs and the dataset we constructed. Therefore, in addition to querying the LVLM locations about images, we also incorporated real-time image search results from Google to provide the model with more comprehensive information. These results from Internet retrievals are incorporated into the chain-of-thoughts (CoT) [47] of LVLMs as external knowledge. At this time, models with larger parameters, such as llava–1.6–34b, demonstrate superior reasoning abilities compared to smaller models (7b or 13b). In addition, closed-source large models also show more consistent performance than their open-source counterparts and are more adept at analyzing and utilizing external knowledge for accurate inferences. Compared to all single LVLMs, our proposed LVLM agent framework surpasses all single LVLM baselines in accuracy. This improvement confirms the effectiveness of different LVLMs collaborating by engaging in discussions and analyzing various types of images, thus producing more precise results.

Table 2: Results of different agent frameworks without web searching.

| Framework | LLM-Blender | PHP | Reflexion | LLM Debate | DyLAN | **smileGeo** |
|---|---|---|---|---|---|---|
| Sturcture |  |  |  |  |  |  |
| Acc ↑ | 55.7802% | 60.9809% | 62.3412% | 57.0119% | 62.8187% | **63.2730%** |
| Tks ↓ | 23,662 | 154,520 | 109,524 | 260,756 | 159,320 | **18,876** |

'Acc' stands for the accuracy of the framework;
'Tks' means the average tokens a framework costs per query (including image tokens).

Secondly, the comparative results across various LLM/LVLM agent frameworks are presented in Table 2. It is evident that the majority of LLM/LVLM agent frameworks surpass individual LVLMs in terms of geo-localization accuracy. This improvement can primarily be attributed to the ability to integrate knowledge from multiple LVLM agents, thereby enhancing the overall precision of these frameworks. However, LLM-Blender and LLM Debate exhibit lower accuracy due to statements of some agents misleading others during discussions, which impedes the generation of correct outcomes. Our framework, smileGeo, guarantees the highest accuracy while being able to accomplish the geo-localization task with the lowest token costs. The average number of tokens our framework spent per query is 18,876, and it is less than the computational overhead of LLM-Blender (23,662), which has the simplest agent framework structure but the lowest accuracy among all baselines. This is mainly due to a 'small' GNN-based dynamic learning model being deployed for agent selection stages and significantly reducing unnecessary discussions among agents.

Finally, Table 3 presents the comparison between the proposed framework and existing image retrieval systems. Our framework, smileGeo, consistently outperforms all other retrieval-based approaches. This superior performance can be attributed to the fact that other image retrieval methods rely on a rich geo-tagged image database. In our test dataset, however, two-thirds of the images

Table 3: Comparison with image retrieval systems.

| | **Natural** | **ManMade** | **Overall** |
|---|---|---|---|
| NetVLAD | 26.5134 | 28.9955 | 28.6047 |
| GeM | 23.1022 | 25.4175 | 25.0749 |
| CosPlace | 28.1688 | 30.2782 | 29.8701 |
| **smileGeo** | **58.6111** | **64.3968** | **63.2730** |

Bold indicates the statistically significant improvements (*i.e.,* two-sided t-test with $p < 0.05$) over the best baseline.

are new and localized in completely different areas from those in the training dataset. This highlights the shortages of conventional database-based retrieval systems due to the limitations of the geo-tagged image databases and demonstrates the effectiveness of our proposed framework in solving open-world geo-localization tasks.

## 4.3 Ablation Study

**Number of Agents**. We further demonstrate the relationships between the number of agents and the framework performance. We conduct experiments in two ways: i) by calling the same closed-source LVLM API (Here, we use Gemini-1.5-pro because it performs best without the help of the Internet) under different prompts (*e.g.,* You are good at recognizing natural attractions; You're a traveler around

Europe) to simulate different agents, and ii) by using different LVLM backbones to represent distinct agents. The results are shown in Figure 2.

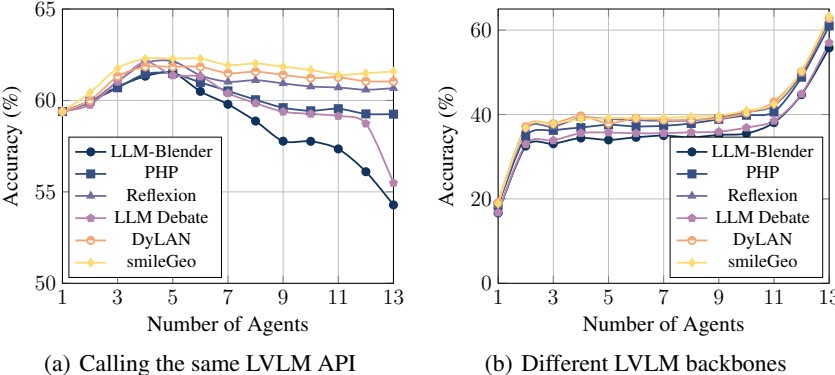

(a) Calling the same LVLM API      (b) Different LVLM backbones

Figure 2: Results of model performance in relation to the number of agents.

As illustrated in Figure 2(a), the framework achieves optimal accuracy with 4 or 5 agents. Beyond this number, the framework's performance begins to deteriorate. This shows that using models with the same knowledge and reasoning capabilities as different agents has limited improvement in the accuracy of the framework. Despite this decline, the performance of frameworks other than LLM-Blender and LLM Debate remains superior to that of a single agent. LLM-Blender and LLM Debate, however, have a significant decrease in model accuracy when the number of agents exceeds 11. This is mainly because both of them involve all LVLMs in every discussion, which suffers from excessive repetitive and redundant discussions. Figure 2(b) reveals that the accuracy of the framework improves with the incorporation of more LVLM backbones, indicating that the diversity of LVLMs can enhance the quality of discussions.

**Hyperparameter $K$ & $R$.** There are two hyperparameters, $K$ and $R$, that need to be pre-defined in the proposed framework: $K$ is the number of agents (answer agents) that respond in each round of discussion, and $R$ is the number of agents (review agents) used to review answers from answer agents. Therefore, we conduct experiments under different combinations of $K \in [1, 8]$ and $R \in [1, 8]$, as shown in Figure 3. The results indicate that optimal performance can be achieved with relatively small values of $K$ or $R$. However, the computational cost, measured in tokens, increases exponentially with higher values of $K$ and $R$. To balance both the efficiency and the accuracy of smileGeo, for

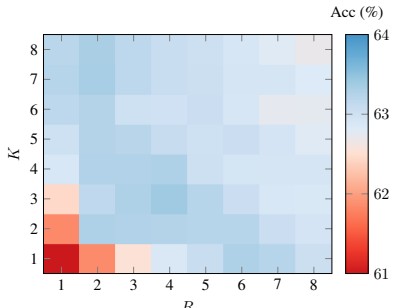

Figure 3: Results under different $K$ and $R$.

the experiments presented in this paper, we set both $K$ and $R$ equal to 2.

## 5 Conclusion

This work introduces a novel LVLM agent framework, smileGeo, specifically designed for geo-localization tasks. Inspired by the review mechanism, it integrates various LVLMs to discuss anonymously and geo-localize images worldwide. Additionally, we have developed a dynamic learning strategy for agent collaboration social networks, electing appropriate agents to geo-localize each image with different characteristics. This enhancement reduces the computational burden associated with collaborative discussions among LVLM agents. Moreover, we have constructed a geo-localization dataset called GeoGlobe and will open-source it. Overall, smileGeo demonstrates significant improvements in geo-localization tasks, achieving superior performance with lower computational demands compared to contemporary state-of-the-art LLM/LVLM agent frameworks.

Looking ahead, we aim to expand the capabilities of smileGeo to incorporate more powerful external tools beyond just web searching. Additionally, we plan to explore extending its application to complex scenarios, such as high-precision global positioning and navigation for robots, laying the cornerstone for exploring LVLM agent collaboration to handle different complex open-world tasks efficiently.

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

## A Notations

We summarize all notations in this paper and list them in Table 4.

Table 4: Notations in this paper.

| Notation | Description |
|---|---|
| $X$ | The image to be recognized. |
| $Y$ ($\hat{Y}$) | The predicted (ground truth of) geospatial location in the natural language form. |
| $\mathcal{G}$ ($\hat{\mathcal{G}}$) | The predicted (ground truth of) LVLM-based agent collaboration social network. |
| $A$ ($\hat{A}$) | The predicted (ground truth of) adjacency matrix of the agent social network. |
| $Lst$ ($\hat{Lst}$) | The predicted (ground truth of) scalar of agent election probability. |
| $\mathcal{V}$ | The set of LLM agents. |
| $\mathcal{E}$ | The set of connections between LLM agents. |
| $N$ | The number of agents. |
| $K$ | The number of agents to be elected as answer agent(s). |
| $R$ | The number of agents to be selected as review agent(s). |
| $L$ | The number of agent discussion rounds. |
| $Z$ | The maximum number of rounds in which answer agents harmonize opinions. |
| $\Theta$ | The learnable parameters of the agent social network learning model. |

## B Dataset Details

The images in this dataset are copyright-free images obtained from the Internet via a crawler. We divide the images into two main categories: man-made landmarks as well as natural attractions. Then, we filter out the data samples that could clearly identify the locations of the landmarks or attractions in the images. As a result, we filter out nearly three hundred thousand data samples, and please refer to Table 5 and Figure 4 for details. Due to the fact that a large number of natural attractions in different geographical regions with high similarity are cleaned, the magnitude of the data related to natural attractions in this dataset is smaller than that of man-made attractions.

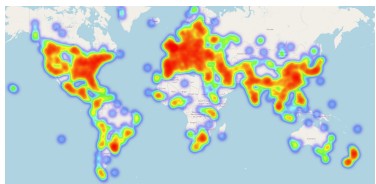

Figure 4: The data distribution around the world.

Table 5: Statistics of the dataset GeoGlobe.

| | Images | Cities | Countries | Attractions |
|---|---|---|---|---|
| Man-made | 253,118 | 2,313 | 143 | 10,492 |
| Natural | 40,087 | 1,044 | 97 | 1,849 |

For an open-world geo-localization task, the relationship between the training and test samples in the experiment could greatly affect the results. We label the training samples as $\mathcal{Z}_{\text{train}}$, and the test sample set as $\mathcal{Z}_{\text{test}}$, and use two metrics, *coverage* as well as *consistency*, to portray this relationship:

$$
\begin{aligned}
coverage &= \frac{\mathcal{Z}_{\text{train}} \cap \mathcal{Z}_{\text{test}}}{\mathcal{Z}_{\text{train}}} \times 100\% \\
consistency &= \frac{\mathcal{Z}_{\text{train}} \cap \mathcal{Z}_{\text{test}}}{\mathcal{Z}_{\text{test}}} \times 100\%
\end{aligned}
\tag{6}
$$

As for the samples in this paper, $coverage \approx 4.6564\%$, and $consistency \approx 33.2957\%$.

## C Implementation Details

In all experiments, we employ a variety of LVLMs, encompassing both open-source and closed-source models, to be agents in the proposed framework. Unless specified otherwise, zero-shot prompting is applied. Each open-source LVLM is deployed on a dedicated A800 (80G) GPU server with 200GB memory. As for each closed-source LVLM, we cost amounting to billions of tokens by calling APIs as specified by the official website. To avoid the context length issue that occurs in some LVLMs, we truncate the context before submitting it to the agent for questions based on the maximum number of

**Algorithm 1** The smileGeo framework

---

**Input:** A set of pre-trained LLMs $\mathcal{V} = \{v_1, v_2, \cdots\}$, the input image $\boldsymbol{X}$, and the ground truth $\hat{\boldsymbol{Y}}$ (if has);

**Output:** The geospatial location $\boldsymbol{Y}$.

    *Initialization Stage:*

1: Initialize (Load) the parameter of the agent selection model: $\Theta$

2: Calculate: $\boldsymbol{A} \leftarrow f(X, \mathcal{V}|\Theta)$

3: Initialize the agent collaboration social network: $\mathcal{G}$

4: Calculate: $\boldsymbol{Lst} \leftarrow f(X, \mathcal{G}|\Theta)$

    *Stage 1:*

5: Elect answer agents: $\mathcal{V}^1 = \{v_a^1, v_b^1, \cdots\} \leftarrow \boldsymbol{Lst}$, where $|\mathcal{V}^1| = K$

6: **for** each answer agent $v^1$ **do**

7:    Obtain the location: $\boldsymbol{Y}_{v^1}^1 \leftarrow \text{Ask}_{v^1}(\boldsymbol{X})$

8:    Get the confidence percentage: $C_{v^1}^1 \leftarrow \text{Ask}_{v^1}(\boldsymbol{X}, \boldsymbol{Y}_{v^1}^1)$

9:    Store the further explanation: $T_{v^1}^1 \leftarrow \text{Ask}_{v^1}(\boldsymbol{X}, \boldsymbol{Y}_{v^1}^1)$

10: **end for**

    *Stage 2:*

11: **for** each selected answer agent $v^1$ **do**

12:    Select the review agents: $\mathcal{V}^2 = \{v_a^2, v_b^2, \cdots\} \leftarrow \text{RandomWalk}_{v^1}(\mathcal{G})$, where $|\mathcal{V}^2| = R$

13:    **for** each review agent $v^2$ **do**

14:        Obtain the comment $T_{v^2}^2 \leftarrow \text{Review}_{v^2}(\boldsymbol{X}, \boldsymbol{Y}_{v^1}^1, C_{v^1}^1)$

15:        Get the confidence percentage: $C_{v^2}^2 \leftarrow \text{Ask}_{v^2}(\boldsymbol{X}, T_{v^2}^2)$

16:    **end for**

17: **end for**

    *Stage 3:*

18: **for** each selected answer agent $v^1$ **do**

19:    Summary the final answer: $\boldsymbol{Y}_{v^1}^3 \leftarrow \text{Summary}_{v^1}(\boldsymbol{Y}_{v^1}^1, C_{v^1}^1, T_{v_1^2}^2, C_{v_1^2}^2, T_{v_2^2}^2, , C_{v_2^2}^2, \cdots)$

20:    Get the final confidence percentage: $C_{v^1}^3 \leftarrow \text{Ask}_{v^1}(\boldsymbol{Y}_{v^1}^1, C_{v^1}^1, T_{v_1^2}^2, C_{v_1^2}^2, T_{v_2^2}^2, , C_{v_2^2}^2, \cdots)$

21: **end for**

22: Generate the final answer: $\boldsymbol{Y} \leftarrow \text{Discussion}_Z(\boldsymbol{Y}_{v_1^1}^3, C_{v_2^1}^3, \boldsymbol{Y}_{v_2^1}^3, C_{v_2^1}^3, \cdots)$

    *The dynamic learning strategy module:*

23: Initialize $\boldsymbol{Lst}^{(0)}, \mathcal{G}^{(0)}$

24: **for** round $l$ in total $L$ rounds **do**

25:    **for** each selected answer agent $v^1$ **do**

26:        Obtain coordinates: $Coors \leftarrow \text{GeoEmb}(\boldsymbol{Y}_{v^1}^3), Coors_{\text{Truth}} \leftarrow \text{GeoEmb}(\boldsymbol{Y}_{\text{Truth}})$

27:        **if** $\text{Dis}(Coors, Coors_{\text{Truth}}) \leq th$ **then**

28:            $\boldsymbol{A}^{(l)} \leftarrow \text{Enhance}(e|e \text{ contains } v^1, e \in \mathcal{E})$

29:            Update $\boldsymbol{Lst}^{(l)}[v^1] = 1$

30:        **else**

31:            $\boldsymbol{A}^{(l)} \leftarrow \text{Weaken}(e|e \text{ contains } v^1, e \in \mathcal{E})$

32:            Update $\boldsymbol{Lst}^{(l)}[v^1] = 0$

33:        **end if**

34:    **end for**

35: **end for**

36: $\hat{\boldsymbol{A}} \approx \boldsymbol{A}^{(L)}, \hat{\boldsymbol{Lst}} \approx \boldsymbol{Lst}^{(L)}$

37: Update: $\Theta \leftarrow Loss(\hat{\boldsymbol{Y}}, \boldsymbol{Y}, \hat{\boldsymbol{A}}, \boldsymbol{A}, \hat{\boldsymbol{Lst}}, \boldsymbol{Lst})$

---

tokens that each agent supports. Besides, noting that images are token consuming, we only keep the freshest response for agent discussions.

The detailed algorithm of smileGeo is illustrated in Algorithm 1. In the initialization stage, we initialize or load the parameters of the agent social network learning model, as delineated in line 1. Next, we treat each LVLM agent as a node, establishing the LVLM agent collaboration social network and computing the adjacency relationships among LVLM agents as well as the probability that each agent is suited for responding to image $\boldsymbol{X}$, as shown in line 2. Then, line 3 initializes the agent

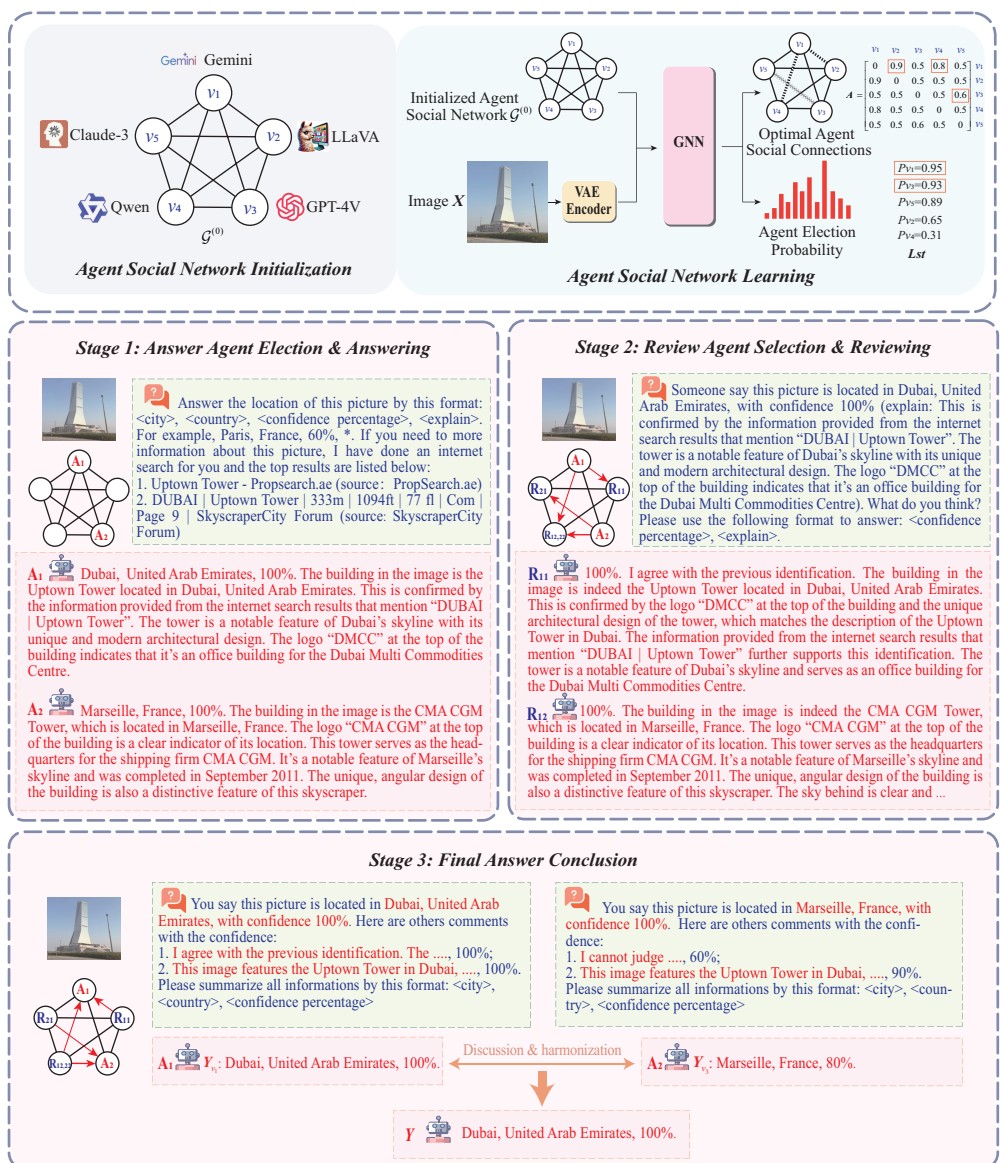

Figure 5: A case study on the geo-localization process via a given image.

collaboration social network and line 4 computes the agent election probability. In Stage 1, line 5 involves electing appropriate answer agents based on the calculated probabilities. Subsequently, lines 6-10 detail the process through which each chosen answer agent formulates their response. Stage 2 begins by employing the random walk algorithm to assign review agents to each answer agent, as depicted in lines 11-12. Lines 13-16 then describe how these review agents generate feedback based on the answers provided. In Stage 3, each answer agent consolidates feedback from their assigned review agents to finalize their response, as illustrated in lines 18-21. Line 22 concludes the final answer with up to $Z$ rounds (we set $Z = 10$ in experiments) of intra-discussion among all answer agents only. The dynamic learning strategy module involves $L$-round (we set $L = 20$ in experiments) comparing the generated answers against the ground truth and updating the connections between the answer and review agents accordingly, as shown in lines 23-36. In line 37, the process concludes with the updating of the learning parameters of the dynamic agent social network learning model.

Here, for the agent social network learning model, we first deflate each image to be recognized to 512x512 pixels and then use the pre-trained VAE model[11] to compress the image again (compression

---

[11]https://huggingface.co/stabilityai/sd-vae-ft-mse

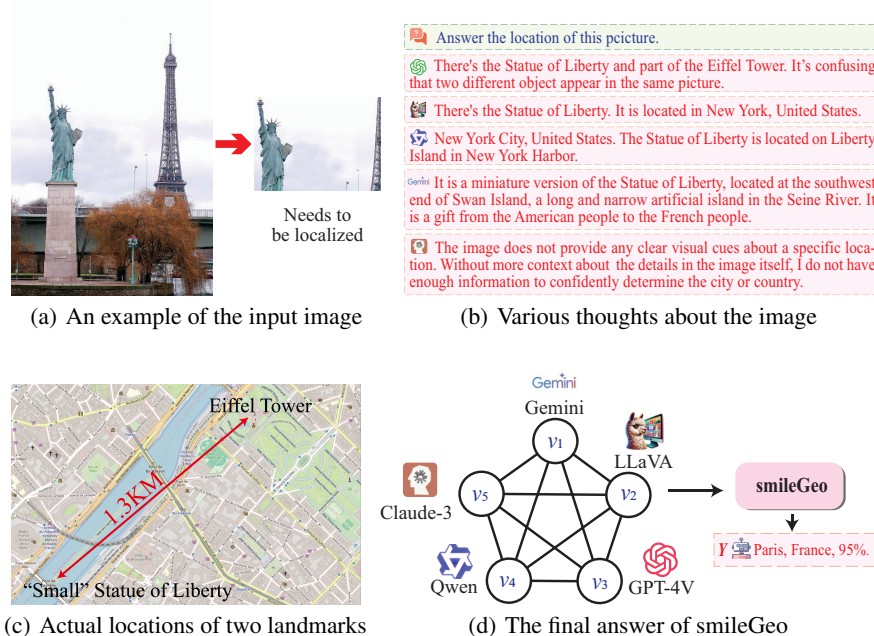

(a) An example of the input image

(b) Various thoughts about the image

(c) Actual locations of two landmarks

(d) The final answer of smileGeo

Figure 6: A case study illustrating the reasoning capabilities of smileGeo.

ratio 1:8) and extract its representations. We define the embedding dimension of the nodes to be 1024 and the hidden layer dimension of the network layer to be 1024. we use Adam as an optimizer for gradient descent with a learning rate of $1e^{-5}$. For each stage of the LVLM agent discussion, we use a uniform template to ask questions to different LVLM agents and ask them to make a response in the specified format. In addition, the performance of our proposed framework is the average of the last 100 epochs in a total training of 2500 epochs.

# D  Additional Experiments

## D.1  Case Study

**Case 1:** In Figure 5, we illustrate the application of smileGeo in a visual geo-localization task. For this demonstration, we randomly select an image from the test dataset and employ five distinct LVLMs: LLaVA, GPT-4V, Claude-3, Gemini, and Qwen. The agent selection model selects two answer agents, as depicted in the top part of the figure. Subsequently, stages 1 through 3 detail the process of generating the accurate geo-location. Initially, only one answer agent provided the correct response. However, after several rounds of discussion, the agent that initially responded incorrectly revised the confidence level of its answer. During the final internal discussion, this agent aligned its response with the correct answer. This outcome validates the efficacy of our proposed framework, demonstrating its ability to integrate the knowledge and reasoning capabilities of different agents to enhance the overall performance of the proposed LVLM agent framework.

**Case 2:** This case study illustrates the need to pinpoint the geographical location of a complete image based on only a portion of it, as demonstrated in 6(a). As illustrated in Figure 6(b), all agents recognized the Statue of Liberty in Figure 6(a), and some identified the presence of part of the Eiffel Tower at the edge of the picture. For instance, GPT-4V concluded that the buildings in these two locations appeared in the same image. However, as is known through the knowledge of other agents (Gemini), a scaled-down version of the Statue of Liberty has been erected on Swan Island, an artificial island in the Seine River in France. By marking both the Eiffel Tower and the island on the Open Street Map (OSM) manually, as shown in Figure 6(c), it is evident that they are merely 1.3 kilometers apart in a straight line. By utilizing the proposed framework, agents discuss and summarize the location depicted in the picture to be Paris, France, as shown in Figure 6(d). Thus, without human intervention, this framework demonstrates the effectiveness of doing geo-localization tasks.

