# OpenReview forum: "Swarm Intelligence in Geo-Localization: A Multi-Agent Large Vision-Language Model Collaborative Framework"
_NeurIPS.cc/2024/Conference — Submitted to NeurIPS 2024_

### Official Review · Reviewer_Jv7J · 2024-07-12

**Soundness:** 3
**Presentation:** 3
**Contribution:** 3
**Rating:** 4
**Confidence:** 4

**Summary:**

Proposed a graph based learnable multi-agent framework. The framework consists of multiple stages : Forwarding: Election (K: Answer agents; R: Reviewer) -> Review -> K Discuss till a final conclusion is reached. Proposed a mechanism to learn the graph connections dynamically.

The major Contributions Introduced in the paper: (A)  A new swarm intelligence geo-local framework smileGeo; (B) Dynamic learning strategy; (C) A new Geo-dataset (test mainly).

**Strengths:**

The major strengths of the proposed smileGeo frameworks are:

 (a) the learnable Graph based communication strategy seems works well empirically. In table 2, authors demonstrated that it helps achieve better acc, but lower average token costs.

(b) The proposed method is also scalable as shown in table 3.

(c) Used attention-based GNN to predict optimal connections and optimal election. Also empirically justified the effectiveness of attention based GNN.

(d) Also constructed Simple rules of updating edges(connections) that works well in practice.

**Weaknesses:**

The major weaknesses are as follows:

(a) Comparisons with baselines seems unfair.

(b) Missing details of the evaluation setup, metrics, etc.

**Questions:**

Here are my questions to the authors:

(a) Table 1 involves comparison between open/closed source single LVLMs with smileGeo-single. However, smileGeo appears to primarily focus on a multi-agent framework, without introducing any new single LVLM architectures.

(b) The comparative results of different agent frameworks without web searching are reported in table 2. How are 'acc' and 'tks' determined for each framework? Which types of LVLMs are employed? Did they aggregate all LVLMs and calculate averages per framework, or utilize the best-performing LVLM specific to each framework? It's important not to unfairly advantage one framework over others by using superior LVLMs.

(c) Question about the comparison with LLM/LVLM-based agent frameworks:

For the integration frameworks you compared in Table 2, what specific LVLMs were integrated within the LLM-Blender, LLM Debate, and smileGeo frameworks? Did you use the same LVLM combinations for the different frameworks in the comparison? Different LVLM combinations may have different underlying behavior on metrics.

**Limitations:**

yes, the authors adequately addressed the limitations.

---

> ### Author Rebuttal · Authors · 2024-08-06
>
> **Q1. Table 1 involves a comparison between open/closed source single LVLMs with smileGeo-single. However, smileGeo appears to primarily focus on a multi-agent framework, without introducing any new single LVLM architectures.**
>
> Thank you for your comments. In fact, Table 1 compares the results directly generated by the LVLM used by each different LLM agent in smileGeo with the results discussed using the smileGeo framework for all LLM agents. The experimental design of Table 1 aims to demonstrate that the LVLM possesses a certain reasoning ability and can infer more accurate results by incorporating external information when discussing with other LVLMs. Additionally, we have included the comparative results of different agent frameworks in Table 2. We will refine the structure of the experiments to avoid any confusion.
>
> **Q2. The comparative results of different agent frameworks without web searching are reported in Table 2. It's important not to unfairly advantage one framework over others by using superior LVLMs.**
>
> Thank you for your comments. Sorry for the confusion about the experiment settings. To ensure the fairness of experiments, all frameworks utilize the same number and types of LLM agents. The only difference lies in the architectures used for discussion among the LVLMs. We would fill in more details of experiment settings to avoid confusion in the revised version.
>
> **2.1 How are 'acc' and 'tks' determined for each framework?**
>
> Thank you for your questions. We have clarified the meanings of 'acc' and 'tks' below Table 2: 'acc' stands for the accuracy of the framework, while 'tks' refers to the average number of tokens a framework uses per query (including image tokens). Additionally, the calculation of 'acc' is provided in Section 4.1, Evaluation Metrics. The number of tokens is determined by the length of the query sentences and the pixels of the images: the longer the sentence or the higher the image resolution, the more tokens there are.
>
> **2.2 Which types of LVLMs are employed?**
>
> Thank you for your questions. Each framework utilizes all the single LVLMs listed in Table 1 as different LLM agents, engages them in discussions, and summarizes the final results.
>
> **2.3 Did they aggregate all LVLMs and calculate averages per framework, or utilize the best-performing LVLM specific to each framework?**
>
> Thank you for your questions. All frameworks are designed to enable all LVLMs to reach a consensus and provide a unified answer. To prevent situations where a consensus cannot be reached, we implement a majority rule after a certain number of discussion rounds, ensuring a unified answer is recognized by the majority of agents.
>
> **Q3. Question about the comparison with LLM/LVLM-based agent frameworks:**
>
> **3.1 For the integration frameworks you compared in Table 2, what specific LVLMs were integrated within the LLM-Blender, LLM Debate, and smileGeo frameworks?**
>
> Thank you for your questions. All frameworks use the single LVLMs listed in Table 1 as distinct agents, engage them in discussions, and summarize the final results.
>
> **3.2 Did you use the same LVLM combinations for the different frameworks in the comparison? Different LVLM combinations may have different underlying behavior on metrics.**
>
> Thank you for your questions. Yes, we use the same LVLM combinations for the different frameworks in the comparison. We apologize for any confusion caused and will refine the experimental section to ensure a clearer expression.

---

> > ### Author Response · Authors · 2024-08-12
> >
> > As we approach the end of the author-reviewer discussion period, we respectfully wish to check in and ensure that our rebuttal has effectively addressed your concerns regarding our paper. Should you have any remaining questions or need further clarification or additional experimental results, please do not hesitate to let us know. We appreciate the thoughtful reviews and the time you’ve invested in providing us with valuable feedback to improve our work. If you believe that our responses have sufficiently addressed the issues raised, we kindly ask you to consider the possibility of raising the score.

---

> > > ### Comment · Reviewer_Jv7J · 2024-08-13
> > >
> > > I appreciate the authors' response. However, I remain unconvinced by their explanation regarding the implementation of the majority rule after a certain number of discussion rounds when consensus cannot be reached. The method for determining the number of discussion rounds requires further analysis, as it is a crucial aspect of their proposed framework. This situation is likely to occur frequently in practice. Additionally, after reviewing the author's discussion with the reviewer mcak, I agree with most of the concerns raised by the reviewer mcak. I find the authors' response to be unconvincing, especially the motivation and cost issues. Therefore, I also recommend a borderline reject.

---

### Official Review · Reviewer_EWQq · 2024-07-14

**Soundness:** 4
**Presentation:** 3
**Contribution:** 4
**Rating:** 6
**Confidence:** 4

**Summary:**

This works proposes a new visual geo-localization framework with multiple LVLM (Large Vision Language Model) agents. The agents communicate with each other to estimate the geo-location of the input image. A dynamic learning strategy is proposed to optimize the communication patterns among agents to improve efficiency. The method is evaluated on the proposed GeoGlobe dataset.

**Strengths:**

+ The idea of tacking worldwide city-level geo-localization with multiple LVLM agents is very interesting.
+ The result is surprisingly good with zero-shot setting, which is even better than powerful close-source models.
+ Detailed comparison with other agent-based methods is provided. The ablation study on the number of agents is also very detailed.
+ The writing is easy to follow.

**Weaknesses:**

- The authors could make the geo-localization setting more clear in the introduction, for example, the paper focuses on worldwide city-level geo-localization. There are lots of different settings for geo-localization problem and this could be confusing for some researchers.
- This paper provides a comparison with three traditional geo-localization methods, i.e., NetVLAD, GeM, and CosPlace. However, these three methods are either retrieval-based landmark matching methods or fine-grained classification-based place recognition methods. It would be better to provide a direct comparison with worldwide geo-localization method on city-level setting, e.g., [A]. Although I believe LVLM-based method is better at this setting, a comparison can make it more convincing.

[A] Pramanick, Shraman, et al. "Where in the world is this image? transformer-based geo-localization in the wild." European Conference on Computer Vision. Cham: Springer Nature Switzerland, 2022.
- There are only two qualitative results in the appendix. Given that the accuracy is over 60%, it should be easy to find successful and failed cases to demonstrate the actual output cases of the proposed methods. It can also better illustrate how multiple agents help the geo-localization process.
- There are also some existing worldwide geo-localization datasets that could be used for more comprehensive evaluation, e.g., IM2GPS3K, YFCC4K.

**Questions:**

See the weaknesses.

**Limitations:**

The authors mentioned the limitations in the checklist.

---

> ### Author Rebuttal · Authors · 2024-08-06
>
> **Q1. The authors could make the geo-localization setting more clear in the introduction, for example, the paper focuses on worldwide city-level geo-localization. There are lots of different settings for geo-localization problem and this could be confusing for some researchers.**
>
> Thank you for your suggestions. Our task aims to achieve city-level localization for any landmark worldwide, without assuming access to a limitless image database. In the revised version, we will clarify the problem of worldwide city-level geo-localization in the introduction. We will detail the differences, difficulties, and challenges of this task compared to high-precision positioning in a local area. Thank you again for the valuable feedback.
>
> **Q2. This paper provides a comparison with three traditional geo-localization methods, i.e., NetVLAD, GeM, and CosPlace. However, these three methods are either retrieval-based landmark matching methods or fine-grained classification-based place recognition methods. It would be better to provide a direct comparison with worldwide geo-localization method on city-level setting, e.g., [A]. Although I believe LVLM-based method is better at this setting, a comparison can make it more convincing.**
>
> Thank you for your advice. We have added a direct comparison with the model under the same geo-localization settings. The comparative results (accuracy in %, without web searching) on our dataset, GeoGlobe, are shown below:
> ||Natural|ManMade|Overall|
> |:--:|:--:|:--:|:--:|
> |NetVLAD|26.5134|28.9955|28.6047|
> |GeM|23.1022|25.4175|25.0749|
> |CosPlace|28.1688|30.2782|29.8701|
> |TransLocator[A]|26.1776|34.1971|32.6259|
> |**smileGeo**|58.6111|64.3968|63.2730|
>
> Due to the limited rebuttal time, we did our best to train the ViT-based model presented in paper [A]. To ensure fairness, we trained all the models for the same length of time (3 days) using the same hardware configuration, and then used the same test dataset to test each model separately to obtain the results. The results demonstrate that our method significantly outperforms the referenced method.
>
> **Q3. There are only two qualitative results in the appendix. Given that the accuracy is over 60%, it should be easy to find successful and failed cases to demonstrate the actual output cases of the proposed methods. It can also better illustrate how multiple agents help the geo-localization process.**
>
> Thank you for your suggestions. In the revised version, we will supplement the paper with detailed case studies to describe both successful and failed cases.
>
> An example of a successful case is illustrated in Figure 6. While a single LVLM may not directly identify local landmarks, it possesses relevant geographical knowledge about the landmarks. Our framework can stimulate reasoning capabilities among LVLMs, resulting in correct positioning.
>
> Regarding failure cases, we will show, for example, that increasing the number of the same LLM agents in Figure 2 leads to excessive repeated and redundant information in the discussion, negatively affecting the experimental results.
>
> Additionally, we will include cases illustrating that web search results provide extra information to our LLM agent framework. This can achieve better outcomes when our framework does not rely on Internet search tools.
>
> Thank you again for your valuable advice. We will incorporate these changes in the revised version.
>
> **Q4. There are also some existing worldwide geo-localization datasets that could be used for more comprehensive evaluation, e.g., IM2GPS3K, YFCC4K.**
>
> Thank you for your comments. We conducted further experiments on those datasets, and the results are illustrated in the table below (accuracy in %, without web searching). We also include the comparative results of the best single LLM agent, Gemini-1.5-pro, in our framework and TransLocator as illustrated in paper [A].
> || IM2GPS3K | YFCC4K |
> |:--:|:--:|:--:|
> |Gemini-1.5-pro|32.1989|11.0009|
> | TransLocator [A]|31.0978|13.4039|
> |**smileGeo**|35.6690|16.0714|
>
> Although all models, including ours, have generally lower accuracy on these datasets, our method still outperforms the others.
>
> It is worth noting that the YFCC4K and IM2GPS3K datasets do not apply artificial filtering to the images, resulting in ambiguous images with almost no geographical clues, such as food photos and portraits. This issue is consistent with the problem mentioned in Section 4.1 of the paper [A] cited in Question 2 above. Therefore, in our research, we invested significant time and effort to construct a novel dataset, GeoGlobe, to better evaluate the worldwide city-level geo-localization task.

---

> > ### Comment · Reviewer_EWQq · 2024-08-12
> >
> > Thanks for the rebuttal. It addresses the concerns and I will keep the rating.

---

> > > ### Author Response · Authors · 2024-08-12
> > >
> > > Thank you for your positive feedback. We appreciate your time and effort throughout this reviewer-author discussion stage.

---

### Official Review · Reviewer_mcak · 2024-07-22

**Soundness:** 4
**Presentation:** 3
**Contribution:** 3
**Rating:** 4
**Confidence:** 3

**Summary:**

The paper introduces smileGeo, a novel framework for visual geo-localization, which involves identifying the geographic location of an image. The authors argue that while Large Vision-Language Models (LVLMs) show promise in this area, their individual performance is limited. SmileGeo leverages the concept of "swarm intelligence" by enabling multiple LVLMs to collaborate and refine their location predictions through a multi-stage review process. To enhance efficiency, the framework incorporates a dynamic learning strategy that optimizes the selection of LVLMs for each image. Furthermore, the paper introduces "GeoGlobe," a new dataset designed to evaluate visual geo-localization models in open-world scenarios where many images depict locations not seen during training. Experimental results demonstrate that smileGeo outperforms existing single LVLMs and image retrieval methods, highlighting the effectiveness of collaborative learning for visual geo-localization.

**Strengths:**

* The idea of using an ensemble of networks/agents for geolocalization is interesting and novel. The authors propose a graph-based social network to enable collaboration between the agents.
* The ability to search the internet and provide the agents with relevant information is interesting and improves the performance on the task of geolocalization.
* The paper proposes GeoGlobe, a new dataset for benchmarking models on the task of geo-localizing landmarks. The dataset could be utilized in future for other learning based geospatial tasks.

**Weaknesses:**

* The paper only seems to tackle the problem of geolocalizing **landmark images**. While this is a challenging problem, the current literature [1, 2, 3] has already tried to address the problem of geolocalizing arbitrary ground-level images. The latter problem requires learning sophisticated geographic and visual features. I think even searching the internet cannot effectively solve the geolocalization problem for non-landmark images.
* Limited applicability: The framework is built entirely upon the capabilities of different LVLMs (e.g. GPT4, LLaVA, etc). It seems the framework cannot generalize beyond the training data used for training LLMs.
* The work fails to address the practical applications and real-life use cases of the framework. Why do we require such a framework?
* The limitation and failure cases are not adequately mentioned in the paper.

[1] Vivanco Cepeda, Vicente, Gaurav Kumar Nayak, and Mubarak Shah. "Geoclip: Clip-inspired alignment between locations and images for effective worldwide geo-localization." Advances in Neural Information Processing Systems 36 (2023).

[2] Haas, Lukas, Michal Skreta, Silas Alberti, and Chelsea Finn. "Pigeon: Predicting image geolocations." In Proceedings of the IEEE/CVF Conference on Computer Vision and Pattern Recognition, pp. 12893-12902. 2024.

[3] Berton, Gabriele, Carlo Masone, and Barbara Caputo. "Rethinking visual geo-localization for large-scale applications." In Proceedings of the IEEE/CVF Conference on Computer Vision and Pattern Recognition, pp. 4878-4888. 2022.

**Questions:**

* At present, each agent sees the same information. It might be interesting to incorporate different kinds of information that is revealed differently to the agents, such as multi-view images or panorama images.
* How much compute time is used for a single inference run?
* Why does the performance of some LLMs decrease with web searching?

**Limitations:**

Limitations are insufficiently addressed in the paper. The future works mentioned in the conclusion are vague and fail to specify specific future directions for the work.

---

> ### Author Rebuttal · Authors · 2024-08-06
>
> **Q1.The paper only seems to tackle the problem of geolocalizing landmark images. While this is a challenging problem, the current literature [1-3] has already tried to address the problem of geolocalizing arbitrary ground-level images. The latter problem requires learning sophisticated geographic and visual features.**
>
> Thank you for your comments. Geo-localization of ground-level images is indeed challenging and will be a focus of our future work. Our current work primarily addresses the geo-localization of landmark images, which we mentioned in the abstract.
>
> In our work, we tackle worldwide city-level geo-localization without relying on an extensive image database, unlike the methods in previous studies [1-3], which compare similar images from a backend database. Such a database reliance can limit model performance.
>
> Moreover, geo-localization requires understanding complex geographic and visual features. We leverage the knowledge and reasoning abilities of LLM agents to design a new framework. We will clarify it in the introduction and apologize for any confusion.
>
> **Q2.I think even searching the internet cannot effectively solve the geolocalization problem for non-landmark images.**
>
> Thank you for your comments. I agree that it is very difficult to locate ambiguous images, such as selfies. However, as noted in the second sentence of the abstract, the focus of our work is to perform city-level geo-localization of various landmarks (not only famous landmarks) worldwide. Additionally, we manually constructed a geolocation-based dataset, GeoGlobe, to filter out most images whose locations could not be determined.
>
> **Q3.Limited applicability: The framework is built entirely upon the capabilities of different LVLMs. It seems the framework cannot generalize beyond the training data used for training LLMs.**
>
> Thank you for your concerns. A single LVLM indeed faces these challenges, as many large models attempt to consume vast amounts of data for pre-training. Our motivation is to address the biases in pre-trained LVLMs by combining their strengths. We designed smileGeo using various LLM agents, and experiments show that it outperforms any single LVLMs. Additionally, web searching results can provide extra information, making the framework more robust.
>
> **Q4.The work fails to address the practical applications and real-life use cases. Why do we require such a framework?**
>
> Thank you for your questions. Geo-localization holds significant application value. Beyond the applications such as robot navigation mentioned in the introduction, many popular consumer-facing applications also rely on geo-localization technology, as noted in the privacy policies of apps like Twitter. By geo-locating social media pictures posted by users in real time and analyzing their mobility patterns, these applications can offer tourists personalized recommendations for attractions and itinerary planning. Along this line, our framework achieves more accurate geolocation, thereby enhancing the usefulness and precision of such applications. We will refine our explanation in the paper and apologize for any confusion caused.
>
> **Q5.The limitation and failure cases are not adequately mentioned.**
>
> Thank you for your comments. We recognize the limitation mentioned in the second point of the checklist, noting that the framework currently relies solely on Internet search tools. However, we believe the framework has potential beyond this. As stated in the conclusion, "Looking ahead, we aim to expand the capabilities of smileGeo to incorporate more powerful external tools beyond just web searching."
>
> In the revised paper, we will include failure cases in the appendix to provide readers with a better understanding of our model. For instance, we will demonstrate a scenario where increasing the number of identical LLM agents in Figure 2 leads to excessive repetition and redundancy, negatively affecting the results.
>
> **Q6.At present, each agent sees the same information. It might be interesting to incorporate different kinds of information that is revealed differently to the agents.**
>
> Thank you for your suggestions. Providing each LLM agent with a different view of the image is indeed an intriguing idea. However, it has settings that are different from our learning task. Allowing all LLM agents to access the complete image ensures that each agent can fully analyze the data from its unique perspective. Our proposed framework aims to integrate the memory and reasoning capabilities of different LLM agents, leading to improved accuracy of geo-localization tasks.
>
> **Q7.How much compute time is used for a single inference run?**
>
> Thank you for your questions. The 99% Response Time for smileGeo is less than 25 seconds. This efficiency is largely due to the agent selection model within our framework, which minimizes unnecessary question-answering and communication overhead with large models. Additionally, the slowest LLM agent in the framework has an average response time of less than 500ms, and the average latency for API calls within our servers in the data center is within 50ms. We also limit each question to a maximum of 20 rounds of discussion. Therefore, our model is computationally efficient and significantly outperforms other LLM agent-based frameworks. We will add this explanation to the final version.
>
> **Q8.Why does the performance of some LLMs decrease with web searching?**
>
> Thank you for your concerns. Some web search results introduce noise into geo-localization. Our web search primarily relies on the Google search engine, which inherently includes advertising URLs and contents similar to our queries. Models with weaker reasoning abilities are more susceptible to being influenced by this noise. It is consistent with what we highlighted in Section 4.2: "Models with larger parameters demonstrate superior reasoning abilities compared to smaller models". We will include a more detailed explanation of this in the revised version.

---

> > ### Comment · Reviewer_mcak · 2024-08-11
> >
> > I thank the authors for their responses. I have a few clarifying questions:
> >
> > [1] What is so __unique__ about the framework that it could only be employed for geolocalization? I think the proposed framework is very general and not unique to geolocalization.
> >
> > [2] "A single LVLM indeed faces these challenges, as many large models attempt to consume vast amounts of data for pre-training. Our motivation is to address the biases in pre-trained LVLMs by combining their strengths."
> >
> > How do you ensure that the LLM agents have sufficient world knowledge that is relevant for the task of geolocalization? Are there any empirical studies to prove the statement?
> >
> > [3] Regarding practical application: How can the framework be used in robot navigation? Robot navigation requires fast response times for decision-making at each stage. The authors have mentioned that "99% Response Time for smileGeo is less than 25 seconds." This does not make the framework scalable for real-time applications.

---

> > > ### Author Response · Authors · 2024-08-11
> > >
> > > **Response to Q1:**
> > >
> > > Thank you for your questions. Geo-localization is a complex task that requires extensive geospatial knowledge and strong reasoning abilities. LVLMs offer a novel approach to visual geo-localization by leveraging their powerful visual question-answering (VQA) capabilities, eliminating the need for external geo-tagged image records. This motivation led us to design a discussion framework around LVLMs to fully utilize their strengths and achieve better results. Since different LVLMs have different memory and reasoning capabilities for geo-localization tasks, we designed an LLM agent selection module in the proposed framework, which can select the most suitable agents for geo-localization of the target image for discussion, thus improving the efficiency of the framework. When the selected LLM agents cannot reach a high-confidence conclusion, they can autonomously call an internet-based geographic image search tool to supplement it with additional positioning information.
> > >
> > > We also appreciate your acknowledgment of our framework’s potential as a general approach that could be extended to other fields, reinforcing that the underlying concept of the LLM-based discussion framework is versatile. In the revised version, we will mention our intention to explore its application to other areas in future work.
> > >
> > > **Response to Q2:**
> > >
> > > Thank you for your questions. This was the motivation behind our first comparative experiment, where we compared different single LVLMs in Table 1. Even without retrieval assistance, most closed-source large model agents (such as GPT-4V and Gemini-1.5-pro) and some open-source large models (like Qwen-VL) achieved higher experimental accuracy than some image retrieval-based methods (as shown in Table 3). This demonstrates that LVLMs inherently possess the ability to analyze and process geo-location data, as well as the capacity to retain geo-location knowledge.
> > >
> > > Furthermore, our framework allows LVLM agents to search the internet and obtain sufficient world knowledge directly. As mentioned in Section 4.2, "models with larger parameters, such as llava–1.6–34b, demonstrate superior reasoning abilities compared to smaller models," leading to significant improvements in accuracy and outperforming traditional retrieval-based geo-localization methods.
> > >
> > > These experiments confirm that LLM agents, particularly closed-source large models and LVLMs with larger parameters, exhibit strong memory and reasoning capabilities for geo-localization tasks, both independently and with additional geo-location information.
> > >
> > > Additionally, there are also many reports verifying that using LVLMs for geo-tagging has gained widespread acceptance; please check the following links [1-3].
> > >
> > > [1] https://x.com/itsandrewgao/status/1785827031131001243
> > >
> > > [2] https://lingoport.com/i18n-term/llm/
> > >
> > > [3] https://www.assemblyai.com/blog/llm-use-cases/
> > >
> > > **Response to Q3:**
> > >
> > > Thank you for your concerns. In this paper, we propose a framework that effectively addresses the geo-localization task, which could be a critical component of robot navigation. Robot navigation typically involves many stages, such as localization, trajectory planning, and execution of the planned route. Tasks like localization and path planning often prioritize accuracy over real-time processing, especially in city-level navigation, where incorrect trajectory planning can lead to significant resource wastage.
> > >
> > > Several studies [1][2] utilize LMM/LVLM agents for UAV dispatching, a specific aspect of robot navigation. While inference using LMM/LVLM is known to be very time-consuming, the successful application of methods in these studies indicates promising prospects for LMM/LVLM-based geo-localization. Additionally, we believe that as LLM technology advances—through methods like quantization, distillation of open-source LLM agents, and calling of more lightweight and faster closed-source LLM agents (e.g., GPT-4o-mini)—our proposed framework will soon be capable of real-time responses. We will include this prospect in the revised paper.
> > >
> > > [1] Liu, S., Zhang, H., Qi, Y., Wang, P., Zhang, Y., & Wu, Q. (2023). Aerialvln: Vision-and-language navigation for uavs. In Proceedings of the IEEE/CVF International Conference on Computer Vision (pp. 15384-15394).
> > >
> > > [2] Zhao, H., Pan, F., Ping, H., & Zhou, Y. (2023). Agent as Cerebrum, Controller as Cerebellum: Implementing an Embodied LMM-based Agent on Drones. arXiv preprint arXiv:2311.15033.

---

> > > > ### Comment · Reviewer_mcak · 2024-08-12
> > > >
> > > > I thank the authors for providing additional clarifications. However, after reading the responses, I am still unsure about the motivation for using such a general framework for the task of geolocalization. The response time of 25 seconds seems significant and the dependence of the framework on closed-sourced LLMs raises questions about whether the framework is cost-effective. Hence, I shall keep my rating unchanged.

---

### Decision · Program_Chairs · 2024-09-25

**Decision:**

Reject

**Comment:**

The paper received final ratings of borderline reject, weak accept, and borderline reject. The reviewers indicated general enthusiasm for the high-level approach but expressed concerns about the appropriateness of the model across various task definitions (robot navigation vs general image geolocalization vs landmark image geolocalization), missing comparison to important previous work, insufficient qualitative examples, and limited discussion/analysis of failure cases.

The AC carefully read the paper, the reviews, the rebuttals, and subsequent discussions. The following are the major factors in recommending rejection:
- while the new dataset is potentially interesting, there needs to be more clarity about the construction process. For example, the supplemental states, "we filter out the data samples that could clearly identify the locations of the landmarks or attractions". This seems to indicate that samples were removed that had clear indications of the location, which seems to disagree with statements in the rebuttal that samples were filtered that weren't location discriminative. Regardless, the exact procedure for this filtering should be clarified.

- there are existing datasets for global scale geolocalization and this work seems to dismiss them because they include some challenging images. This work would have been much stronger if it had directly compared to results in recent geolocalization papers that use the IM2GPS3K and YFCC4K datasets. Some attempt was made in the rebuttal phase, but it would have been much better if the original numbers from the other papers had been used rather than a quick attempt to retrain their models.

- for this method to be useful, I would expect it would need to be significantly more accurate than recent work given the high computational complexity and slow run-time relative to methods that are based on image-only processing

- there is also significant potential for data leakage since testing images were drawn from Internet repositories which are also likely to have been used as training data for the LVLMs.

Given these factors, this paper needs an additional round of reviews before it can be accepted.